# The rediscovered motor-related area 55b emerges as a core hub of music perception

Tali Siman-Tov [1,2✉], Carlos R. Gordon[2,3], Netanell Avisdris [1,4,7], Ofir Shany[1,7], Avigail Lerner[1,3], Omer Shuster[5], Roni Y. Granot[5,8] & Talma Hendler [1,2,3,6,8]

Passive listening to music, without sound production or evident movement, is long known to activate motor control regions. Nevertheless, the exact neuroanatomical correlates of the auditory-motor association and its underlying neural mechanisms have not been fully determined. Here, based on a NeuroSynth meta-analysis and three original fMRI paradigms of music perception, we show that the long-ignored pre-motor region, area 55b, an anatomically unique and functionally intriguing region, is a core hub of music perception. Moreover, results of a brain-behavior correlation analysis implicate neural entrainment as the underlying mechanism of area 55b's contribution to music perception. In view of the current results and prior literature, area 55b is proposed as a keystone of sensorimotor integration, a fundamental brain machinery underlying simple to hierarchically complex behaviors. Refining the neuroanatomical and physiological understanding of sensorimotor integration is expected to have a major impact on various fields, from brain disorders to artificial general intelligence.

[1] Sagol Brain Institute Tel Aviv, Wohl Institute for Advanced Imaging, Tel Aviv Sourasky Medical Center, Tel Aviv, Israel. [2] Sackler School of Medicine, Tel Aviv University, Tel Aviv, Israel. [3] Sagol school of Neuroscience, Tel Aviv University, Tel Aviv, Israel. [4] School of Computer Science and Engineering, The Hebrew University of Jerusalem, Jerusalem, Israel. [5] Musicology Department, The Hebrew University of Jerusalem, Jerusalem, Israel. [6] School of Psychological Sciences, Tel Aviv University, Tel Aviv, Israel. [7]These authors contributed equally: Netanell Avisdris, Ofir Shany. [8]These authors jointly supervised this work: Roni Y. Granot, Talma Hendler. ✉email: talisimantov@mail.tau.ac.il

Motor brain regions have long been noticed to be activated during listening to music, even in the absence of any overt bodily movement[1–3]. Imaging studies over the last decades have confirmed repeatedly the involvement of motor control regions, particularly the premotor cortices, basal ganglia, and the cerebellum, in tasks related to rhythm perception[4,5], but importantly, also in tasks evaluating non-rhythmic aspects of music, such as melody and harmony[6,7]. A controversy still exists as to whether entirely passive listening to music activates regions outside the auditory system. Several studies have argued that motor regions are recruited only when a cognitive task is performed concurrently (e.g., anticipation or discrimination)[6,8]; whereas a few others reported motor activations even during purely passive listening[9,10], supporting the view that passive listening is never wholly passive[11].

Why motor-related brain regions are so closely associated with music perception has long fascinated the music cognition community. Among the most cited theories proposed to account for this phenomenon are the Action Simulation for Auditory Prediction (ASAP) thesis by Aniruddh Patel and John Iversen[5], the theory of sensorimotor control through the dorsal auditory stream by Josef Rauscheker[12], and the Habitual Pragmatic Event Map framework by Ricarda Schubotz[13]. Common to all the above is the general understanding that, contrary to traditional views, perception is not a process of passive registration of external stimuli, but rather an active process of exploration and inference. Perception and action are considered tightly intertwined within a complex operation involving predictive mechanisms. Whenever perceiving a stimulus, simulation of the action which may have generated it helps improve perception, and whenever executing an action, the prediction of its sensory consequences optimizes its performance. This neuroscientific paradigm has been empowered over the last two decades through the predictive coding / active inference thesis advanced by Karl Friston and his colleagues[14–16]. In brief, predictive coding maintains that the brain is a hierarchical statistical machine that constantly and iteratively generates top-down predictions and concurrently strives to minimize bottom-up prediction errors. Within the active inference framework, perception-action coupling is critical for prediction error minimization, the presumed underlying computational principle of all brain activity. Music and speech are two domains that best exemplify how perception-action coupling and prediction at multiple levels enhance performance and promote the establishment of higher-level cognition.

Among the theories mentioned above, more specific to music is the ASAP hypothesis, which posits that while listening to music, neural substrates involved in simulation of body movements align their activity with the musical beat (the perceived periodic pulse underlying music), to facilitates auditory temporal prediction and perception[5]. Beat is a fundamental component of music in every human culture[17] and the prediction of beat timings is thought to support music perception[5]. The ASAP suggests that existing timing-based cortical mechanisms in motor control regions are utilized for this predictive process[5].

Within ASAP, the proposed underlying mechanism for the auditory-motor interaction is entrainment, the gradual adjustment and consequent synchronization of two or more independent oscillators[18]. Evidence for behavioral entrainment to the musical beat, i.e., gradual adjustment of an internal body rhythm to an external rhythm (here later referred to as 'rhythmic entrainment') has been reported at multiple levels including perceptual, attentional, autonomic physiological and motor[18,19]. Rhythmic entrainment is considered a robust and intuitive phenomenon central to music perception and production[18]. At the neurophysiological level, brain-to-stimulus neural entrainment has been proposed to mediate rhythmic entrainment[20]; yet,

evidence for brain-to-brain auditory-to-motor entrainment is scarce. Consistent with the ASAP, it has been argued that brain-to-stimulus entrainment facilitates not only temporal or beat perception, but also sensory perception in general, possibly through fluctuations of attention[21–23]. Support for the association between entrainment and temporal prediction, implicated by the ASAP, can be found in the phenomenon termed negative asynchrony; when human subjects are asked to synchronize a motor action to the musical beat (usually finger tapping in sensorimotor synchronization tasks), they tend to anticipate it and act just prior to the beat, usually in the range of tens of milliseconds[5,24].

Interestingly, although entrainment in its general sense is ubiquitous in nature, negative asynchrony during synchronization to paced stimuli appears relatively unique to humans[25,26]. This uniqueness has led to the proposal that rhythmic entrainment relies on auditory-motor pathways which have been evolved to serve vocal learning, the ability to learn to produce novel vocalizations based on auditory experience, imitation and feedback. Vocal learning is relatively rare in the animal kingdom, it has only been reported in a small group of species, including songbirds, parrots, hummingbirds, bats, cetaceans, pinnipeds, elephants, and humans[25], and indeed, several of these species demonstrate relatively high rhythmic entrainment skills[25]. Notably, brain regions related to vocal learning, i.e., the dorsal auditory pathway and motor planning regions have been linked to rhythmic entrainment[5,24], however, its exact neuroanatomical infrastructure has not been fully elucidated.

Among the motor-related regions implicated in music perception, a lateral frontal area stands out in its consistency. It has been identified as a premotor[1,9,27,28], primary motor[1,29], or motor-premotor[6] region. Here we first suggest it nicely fits area 55b, a forgotten cortical region, which despite being reported by Hopf in 1956, only recently has gained recognition through the Human Connectome Project's multi-modal cortical parcellation, HCP-MMP1[30]. Located at the posterior end of the middle frontal gyrus, area 55b is bounded by the primary motor cortex posteriorly, the frontal eye field superiorly, the premotor eye field and ventral premotor cortex inferiorly and the prefrontal cortex anteriorly. The differentiation of area 55b from its neighbors mostly relied on relatively low intracortical myelin content, contrasted with high and moderate myelination of the primary motor cortex and frontal/premotor eye fields, respectively[30]. Beyond assisting in anatomical delineation, sparse myelination may point to higher functioning of area 55b than expected from its pre-motoric location[31].

Glasser and his colleagues argued that area 55b (particularly on the left hemisphere) is involved in language processing, based upon its activation during a functional magnetic resonance imaging (fMRI) language task (listening to a story vs. baseline) and its functional connectivity with the left inferior frontal gyrus (Broca's area)[30]. Since then, a few additional studies have linked area 55b to language, mainly to language production[32–34]. Interestingly, area 55b has also been associated with negative motor responses[35], which though controversial, have been implicated in inhibition of self-movement during action observation, and hence, may colocalize with mirror neurons[35,36]. In summary, albeit recently highlighted as a potential key cortical region, the exact function of area 55b is still to be defined.

The current study has aimed to: (1) tie area 55b, a motor-related region, to music processing in general; (2) substantiate the involvement of area 55b in music perception; and (3) offer a neural mechanism to underlie area 55b's contribution to music perception. We hypothesized that if area 55b, a premotor region, is involved in passive music perception, it may have a role in auditory-motor integration, and that entrainment may serve as the underlying mechanism for this integration. For aim 1, we

used a meta-analysis of music imaging studies; for aim 2, three in-house fMRI paradigms of rhythm, melody, and harmony perception were applied; and for aim 3, results of an accompanying behavioral study were subjected to inter-individual brain-behavior analysis.

Our results indicate that area 55b, particularly on the right hemisphere, is a key hub of music perception. This small, well-defined, so-far overlooked brain region emerged as the most strongly activated cortical area outside the auditory region in various tasks of music perception. Furthermore, activity within the right area 55b was shown to correlate, in a highly selective manner, with indices of rhythmic entrainment. Overall, the current findings support a role for area 55b in sensorimotor integration possibly through neural entrainment.

## Results

**The right area 55b, a premotor region, is a hub of music processing**. To generally link area 55b with music processing, we first used the NeuroSynth online database (https://neurosynth.org/[37]) to meta-analyze 163 neuroimaging studies mentioning the term music. As expected from the meta-analytic procedure used by NeuroSynth (see "Methods"), the included neuroimaging studies dealt with different aspects of music: perception (107 studies), production (8), intervention/training (7), imagery (4), and various combinations of the above (14). The remaining 23 studies did not focus on music processing according to our screening (Supplementary Data 1). The thresholded meta-analytic activation map downloaded from neurosynth.org was first overlaid on a T1 Montreal Neurological Institute (MNI) template. As shown in Fig. 1a, a prominent lateral frontal activation was noticed, consistent with previous literature and the hypotheses of the current study. This activation was more pronounced on the right hemisphere. The meta-analytic map was then projected on a volumetric version of the HCP-MMP1 parcellation. As displayed in Fig. 1b–d, an overlap was observed between the right lateral frontal activation of the meta-analytic map and the right area 55b of the HCP-MMP1. The activation within the right area 55b was the most robust among cerebral activation foci outside the auditory (transverse and superior temporal/insular/opercular) region.

**The right area 55b is activated during passive music listening**. Since only 65% of the studies included in the NeuroSynth meta-analysis focused on music perception, and some of the remaining might have directly activated motor brain regions (music production studies), to establish the role of area 55b in music perception, we introduce results of a recent fMRI study from our laboratory, designed to explore the brain underpinnings of implicit prediction accompanying music listening. Musical excerpts manipulating rhythmic, melodic, and harmonic expectations were presented to 71 healthy non-musicians (mean age: 26.8 years, 41 female) in three separate block-design paradigms. Each paradigm was composed of 16 experimental blocks (block duration: 15–18 s) of either rhythmic phrases (Rhythm paradigm), instrumental melodies (Melody paradigm), or harmonic progressions (Harmony paradigm) at one of four levels of musical complexity (i.e., prediction violation). Rhythmic complexity was defined based on syncopation (violation of metric expectations), melodic complexity was determined through computation of pitch information content and harmonic complexity relied on chord functions in tonal music (see "Methods" and Supplementary Figs. 1–3). Subjects were instructed to lie still, fixate on a central cross and listen to the musical excerpts without performing any particular task. For the purpose of the current study, statistical parametric maps of each paradigm were computed for the contrast of all conditions versus baseline (i.e., across complexity levels), to yield maximal activations associated with

musical perception. The statistical parametric maps were overlaid on a volumetric version of the HCP-MMP1 and mean activity per parcel was calculated for each map. Figure 2 demonstrates the top 10% most strongly activated regions among the 360 parcels of the HCP-MMP1 for each activity map. Ranking the 360 regions based on mean activity per parcel, area 55b on the right hemisphere (R55b) was found at location 24, 32, and 29 in the Rhythm, Melody, and Harmony paradigms, respectively, secondary only to peri-auditory regions. The only additional extra-auditory region among the 36 most activated parcels was the right frontal eye field (RFEF), which was less prominently activated relative to its neighbor, the R55b (Rhythm paradigm, location 32; Melody paradigm, not included; Harmony paradigm, location 35). Summarizing, during passive listening to music the R55b was found to be the most strongly activated cerebral parcel outside the auditory area, irrespective of the aspect of music being presented (i.e., rhythm, melody, or harmony).

**Activity within the right 55b correlates with metrics of rhythmic entrainment**. Among the 71 volunteers who participated in the fMRI study described above, 59 subjects (mean age: 26.3 years, 35 female) also underwent a behavioral assessment. In most cases, the behavioral session took place a few days prior to the imaging session, and included, among others, a sensorimotor synchronization task. While listening to short musical excerpts adopted from the Harvard Beat Assessment Test[38], participants were instructed to tap their right index finger in synchrony with the beat. Synchronization accuracy and consistency measures were calculated using linear and circular methods[38,39]. For the evaluation of synchronization accuracy, we used the absolute (not signed) linear distance in time between subject's taps and the musical beat (absolute asynchrony (AA))[39]. The lower the AA, the more accurate was the subject's performance. For the assessment of synchronization consistency, we used two circular metrics: (1) length of resultant vector (LRV), which represents the distribution of the relative phase angles and ranges from 0 to 1, where 1 denotes perfect concordance of angles (LRV values were submitted to logit transformation to reduce data skewness); and (2) entropy of relative-phase distribution (ENT), which like LRV represents circular spread and ranges from 0 to 1, yet, it uses Shannon entropy, to allow differentiation of random phase distribution from the mixture of in-phase and anti-phase locking[38].

To explore whether these measures of rhythmic entrainment are related to brain activity within the R55b, we first computed a correlation matrix between the behavioral (three indices of rhythmic entrainment) and brain (strength of activation within the R55b during each of the four conditions (complexity levels) of the three musical paradigms) variables. As shown in Fig. 3, a positive correlation was found between performance on the sensorimotor synchronization task and activity within the R55b. Overall, the association between indices of rhythmic entrainment and R55b activity was most significant for the Rhythm paradigm. Variation was noted among the four experimental conditions of each paradigm, such that maximum correlation was found for the third condition of the Rhythm paradigm (moderately syncopated sequences), second condition of the Melody paradigm (relatively simple melodic lines extracted from Mozart's string quartets) and the first condition of the Harmony paradigm (harmonic progressions ending with regular cadence) (Fig. 3). Notably, the pattern of variation in brain-behavior correlation among the experimental conditions overlapped with the variation in emotional ratings provided by subjects following scanning. That is, the experimental conditions showing maximum brain-behavior correlation were also those associated with maximal pleasure (Supplementary Table 1 and Supplementary Fig. 4).

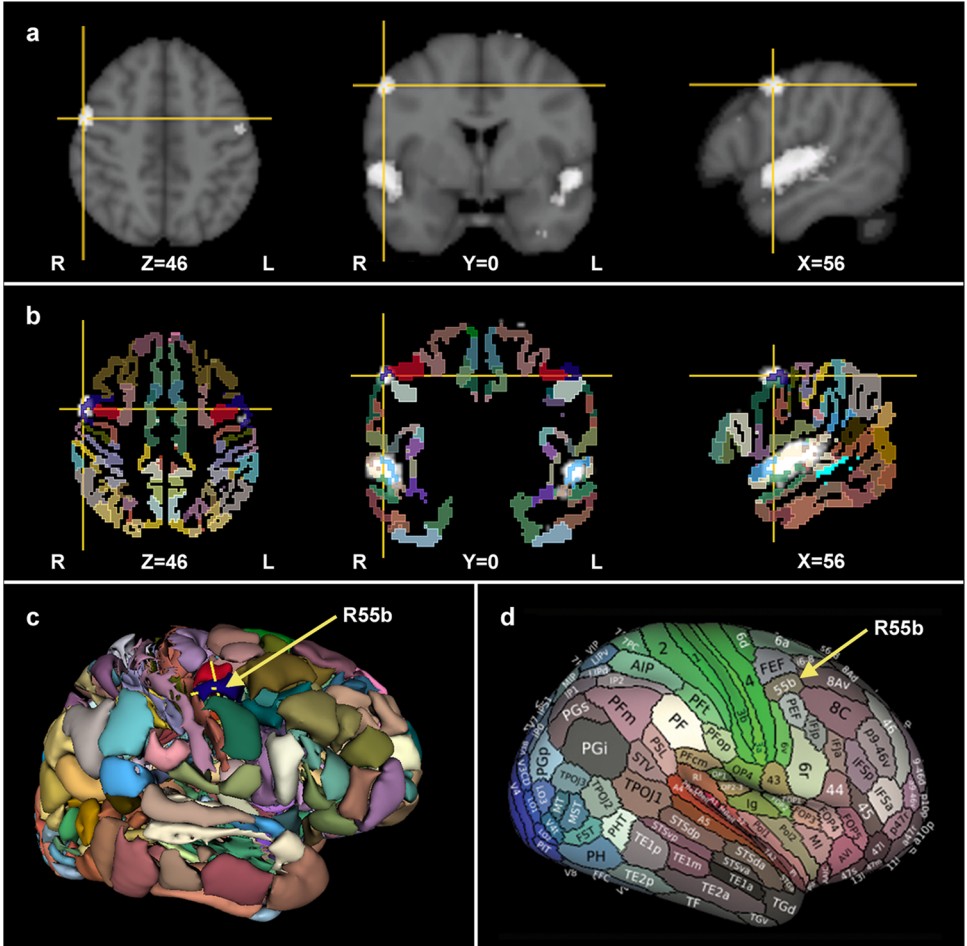

**Fig. 1 Meta-analysis of studies using the term music. a** The NeuroSynth Music meta-analysis map (163 imaging studies, false discovery rate (FDR)-corrected, $p < 0.01$) overlaid on a T1 brain template. The yellow crosshair indicates the peak cerebral activation outside the auditory region (MNI coordinates: 56, 0, 46). **b** The NeuroSynth Music meta-analysis map overlaid on a volumetric version of the HCP-MMP1 (MNI space, ICBM 152 non-linear 6th generation). The largest cerebral extra-auditory activation focus of the meta-analysis (white cluster indicated by the crosshair) overlaps right area 55b (R55b, blue parcel). **c** 3D view of the volumetric version of the HCP-MMP1 parcellation. The yellow crosshair marks the voxel of maximal extra-auditory activation included within the R55b. **d** HCP-MMP1 parcellation of the right hemisphere displayed on inflated surface, as provided by Glasser et al.[30] (downloaded from BALSA, https://balsa.wustl.edu/sceneFile/Zvk4).

To control for potential confounders in the relationship between activity within the R55b and synchronization to the beat, multiple linear regression analyses were performed, using the metrics of rhythmic entrainment as dependent variables, and R55b activity, age, gender, and musical education as independent variables (Table 1). This analysis was conducted for each paradigm separately; mean contrast estimate values were extracted from the R55b for the condition showing maximum brain-behavior correlation in each paradigm. Most significant results were documented for the third condition of the rhythm paradigm (Table 1 and Fig. 4); activity within the R55b while listening to moderately syncopated sequences significantly predicted the performance on a rhythmic entrainment task (AA: $\beta = -15.030$, $p < 0.0003$, LRV-logit: $\beta = 0.518$, $p < 0.00009$, ENT: $\beta = 0.046$, $p < 0.0001$). To rule out the possibility that the above results were affected by computational inaccuracies related to the conversion of the HCP-MMP1 parcellation from surface- to volume-based coordination system, we repeated the multiple linear regression analysis using brain data derived from a 6 mm sphere centered on the extra-auditory peak cerebral activation during the Rhythm paradigm (instead of the R55b parcel). Similar results were obtained (AA: $\beta = -13.926$, $p < 0.001$, LRV-logit: $\beta = 0.428$, $p < 0.001$, ENT: $\beta = 0.039$, $p < 0.001$), as detailed in Supplementary Table 2.

Last, a bidirectional stepwise multiple linear regression was used to examine whether activity in other brain regions significantly activated during passive listening to rhythmic patterns is also associated with indices of rhythmic entrainment. For this analysis, we used the 36 HCP-MMP1 parcels, which showed maximal activation during the Rhythm paradigm (Fig. 2). Values of mean activity within each of the 36 regions (during the third condition of the Rhythm paradigm) served as independent variables, as were age, gender, and musical education. The dependent variables for this analysis were, as before, the three metrics of rhythmic entrainment (AA, LRV-logit, and ENT). The results showed that the optimized model in all three analyses included the R55b activity as the most significant explanatory variable (Tables 2 and 3). In the case of AA, activity within no region other than the R55b was entered to the model. In the case of LRV-logit and ENT, activity within the left perisylvian language area (PSL) and the left frontal operculum 1 region (FOP1) were also included; however, activity within the R55b explained a significant amount of variance in the dependent variables, well beyond that accounted for by the two additional regions. Moreover, FOP1 activity was negatively correlated with the indices of rhythmic entrainment (Tables 2 and 3). In sum, among the 36 brain regions significantly activated by rhythmic sequences, the R55b best accounted for variance in measures of rhythmic entrainment.

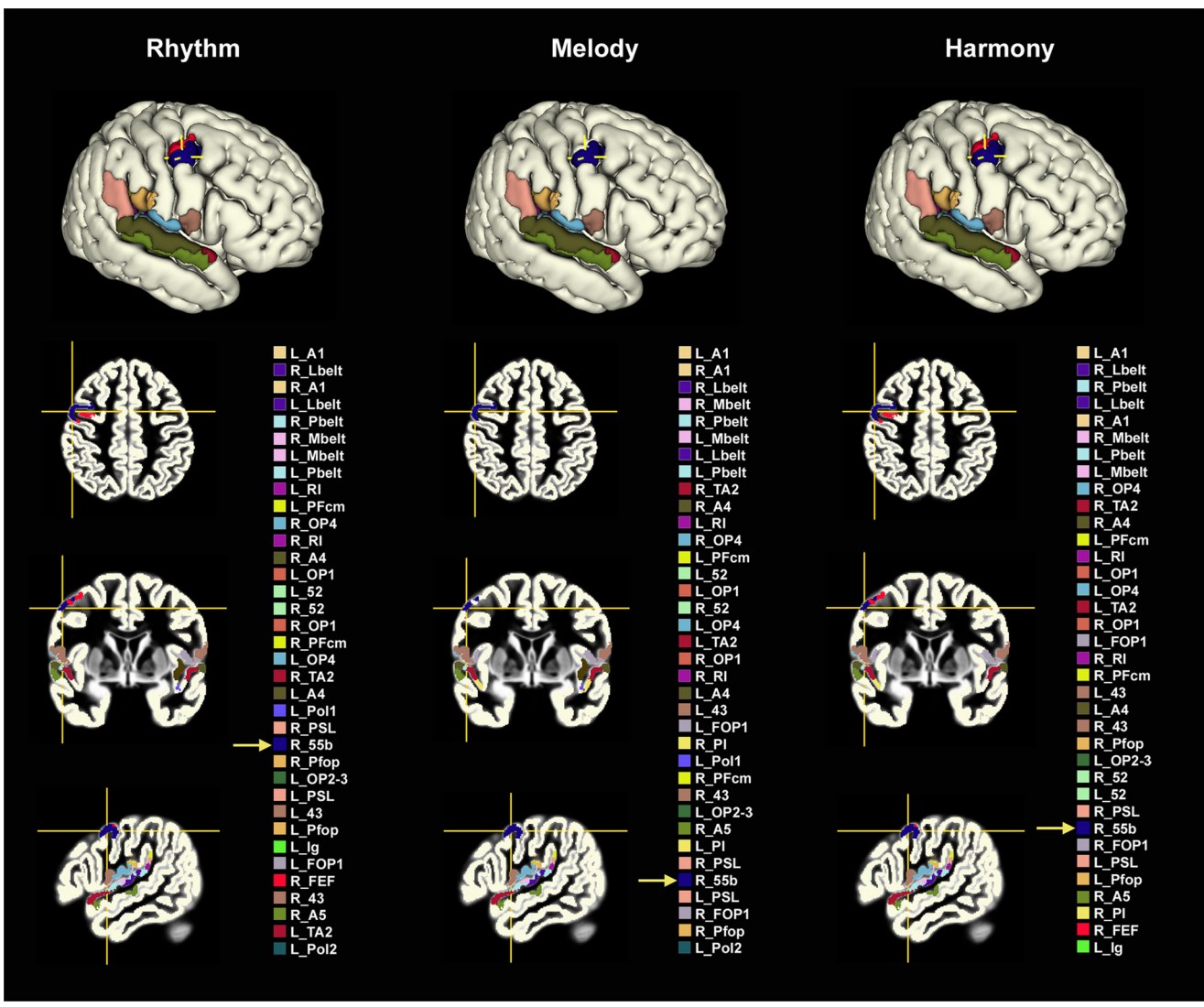

**Fig. 2 Results of three fMRI studies of music perception, introducing HCP-MMP1's most activated parcels during passive listening.** Statistical parametric maps (family-wise error (FWE)-corrected, $p < 0.05$) of three musical paradigms involving passive listening to either rhythmic phrases (Rhythm paradigm, $n = 67$), instrumental melodies (Melody paradigm, $n = 66$) or harmonic progressions (Harmony paradigm, $n = 65$) were overlaid on a volumetric version of the HCP-MMP1 parcellation (MNI space, ICBM 152 2009c non-linear asymmetric). Mean activity per parcel was calculated for each map. The top 10% most strongly activated regions were colored in each parcellation and are listed below the 3D rendering of the right hemisphere, right to the axial, coronal, and sagittal slices at the level of R55b (blue parcel). R55b emerged as the most robust cerebral parcel outside the auditory region. MNI coordinates of the peak cerebral extra-auditory activation: Rhythm paradigm: 52, −2, 52; Melody paradigm: 52, −2, 50; Harmony paradigm: 54, 0, 54. The nearby right frontal eye field (RFEF, red parcel) appeared in two of the three paradigms, its intensity level was lower relative to R55b. See Supplementary Figs. 5–7 for coronal slices covering all 36 regions and Supplementary Data 2 for full names of HCP-MMP1 regions.

## Discussion

In this study, we show that the reintroduced premotor region, named area 55b, particularly on the right hemisphere, is a central hub of music perception. This conclusion, drawn from results of a NeuroSynth meta-analysis and three in-house fMRI paradigms, of rhythmic, melodic, and harmonic perception, endorses the view that passive music listening is never entirely passive, but more importantly, speaks to the involvement of a highly restricted premotor region in this intriguing phenomenon. Furthermore, based on combined behavioral and fMRI data, we propose that area 55b is involved in rhythmic entrainment. While awaiting further behavioral and neurophysiological support, the above findings indirectly suggest that neural entrainment may underlie area 55b's contribution to music perception. In agreement with the ASAP hypothesis which links motor activity with temporal prediction, R55b activation and its association with rhythmic entrainment were more pronounced when listening to rhythms, compared to melodies and harmonic progressions. Nonetheless, the predictive nature of entrainment may implicate its relevance not only to temporal- but also to pitch-related musical elements, as well as to language and other behaviors leaning on complex hierarchical sequences[40]. For now, however, it cannot be ruled out that area 55b activation during listening to melodies and harmonic progressions (in this and other studies) is related to their inevitably associated rhythmic structure. In what follows, we will present evidence linking music perception and auditory-motor integration with neural entrainment, then discuss how area 55b fits into previous anatomical/functional accounts of sensor-imotor association, i.e., the dorsal stream and the mirror neuron system models.

First, it should be noted that the emergence of area 55b as the most robust cortical activation outside the auditory region during

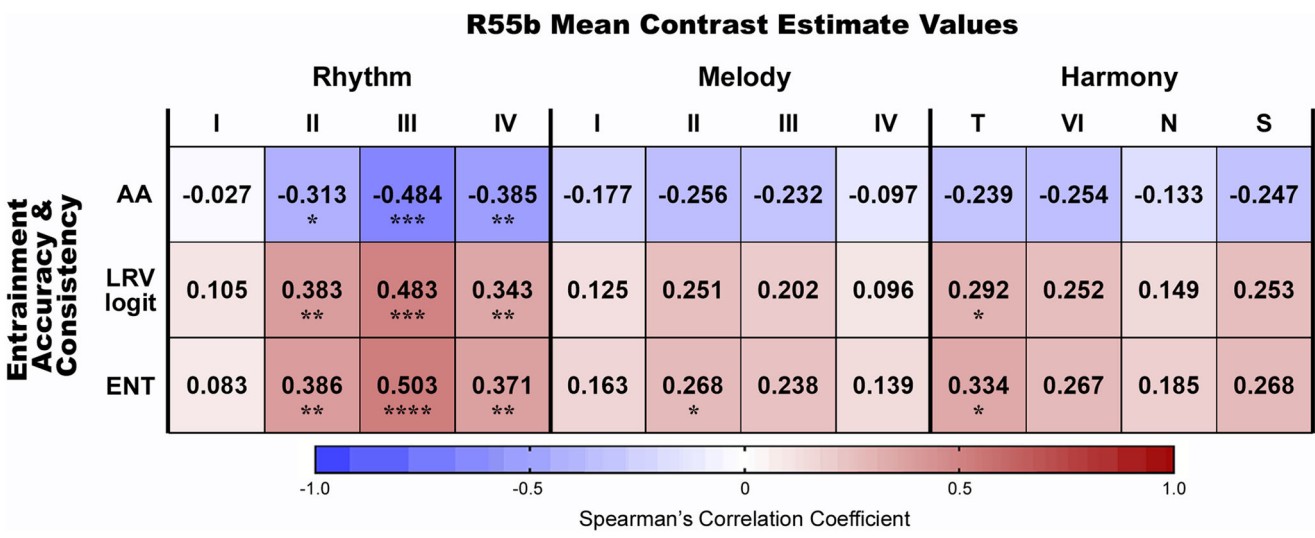

**Fig. 3 Brain-behavior correlation matrix.** Spearman correlation analysis showed positive correlation between accuracy and consistency measures of rhythmic entrainment and activity within the R55b while listening to rhythmic patterns (Rhythm paradigm, $n = 56$), short melodies (Melody paradigm, $n = 58$) and harmonic sequences (Harmony paradigm, $n = 53$). The correlation was most significant for activity values derived from the Rhythm paradigm, and in particular, its third condition (moderately complex rhythms). Less significant results were documented for activity during the second condition of the Melody paradigm (relatively simple Mozart's melodic lines) and the first condition of the Harmony paradigm (regular cadence). For the assessment of synchronization accuracy, we used the absolute values of negative asynchrony (absolute asynchrony (AA)). Lower AA scores indicate higher accuracy. For the assessment of synchronization consistency, we used the circular metrics Length of Resultant Vector (following logit transformation (LRV-logit)) and entropy of relative-phase distribution (ENT). I–IV, complexity levels 1–4, T, Tonic (regular cadence), VI, sixth degree (less regular cadence), N, Neapolitan (irregular cadence), S, scrambled version of the chord progression ending on the tonic. The color bar indicates the Spearman's correlation coefficient, ranging from −1 (blue, strong negative correlation) to +1 (red, strong positive correlation). $*p < 0.05$, $**p < 0.01$, $***p < 0.001$, $****p < 0.0001$.

**Table 1 Multiple linear regression results, R55b association with rhythmic entrainment.**

| | Multiple regression | | | Partial regression coefficients | | | |
|---|---|---|---|---|---|---|---|
| | $R^2$ | *F*-value | *p*-value | Contrast estimate | Age | Gender | MusEdu |
| Rhythm paradigm ($n = 56$) | | | | | | | |
| AA | 0.236 | 3.941 | 0.007** | −15.030*** | −0.238 | −5.181 | −1.139 |
| LRV-logit | 0.308 | 5.671 | 0.001*** | 0.518**** | −0.018 | 0.377 | 0.044 |
| ENT | 0.296 | 5.354 | 0.001*** | 0.046**** | −0.001 | 0.030 | 0.005 |
| Melody paradigm ($n = 58$) | | | | | | | |
| AA | 0.072 | 1.028 | 0.401 | −6.132[a] | −0.023 | −1.176 | −1.034 |
| LRV-logit | 0.144 | 2.227 | 0.078 | 0.220*[b] | −0.025 | 0.211 | 0.043 |
| ENT | 0.124 | 1.880 | 0.128 | 0.019[c] | −0.001 | 0.014 | 0.005 |
| Harmony paradigm ($n = 53$) | | | | | | | |
| AA | 0.068 | 0.877 | 0.485 | −5.157[d] | −0.021 | 3.538 | −1.333 |
| LRV-logit | 0.159 | 2.269 | 0.075 | 0.252[e] | −0.027 | 0.117 | 0.053 |
| ENT | 0.148 | 2.085 | 0.097 | 0.022[f] | −0.001 | 0.006 | 0.006 |

Dependent variables: Absolute asynchrony (AA), length of resultant vector - logit (LRV-logit), entropy of relative-phase distribution (ENT). Independent variable: R55b mean contrast estimate values, age, gender, and musical education (MusEdu). $*p < 0.05$, $**p < 0.01$, $***p < 0.001$, $****p < 0.0001$.
[a]$p = 0.075$, [b]$p = 0.043$, [c]$p = 0.052$, [d]$p = 0.232$, [e]$p = 0.067$, [f]$p = 0.073$.

passive listening to music is in line with several prior meta-analyses of music perception[1,29,41–43]. These studies reported prominent activations in regions that have been identified anatomically as the right precentral or middle frontal gyri, and with our new perspective seem to overlap with the R55b. The emergence of the neighboring RFEF in some of our analyses is not unexpected, considering the high inter-individual anatomical variability described for area 55b[30]. As participants were instructed to fixate on a central cross throughout the experiment this activation is probably not related to eye movements. The right lateralization of area 55b activation observed in our results is consistent with part of the literature on hemispheric dominance of music perception in non-musicians[44,45]. The discussion on music lateralization, and its relations with language lateralization,

is complex and extends beyond the scope of the current study; however, it is interesting to mention that speech envelope tracking, the driving of auditory cortex activity by slow amplitude fluctuations of the speech signal, has been described as right lateralized[46].

An old question is whether neural tracking of speech and music envelopes can be considered a process of entrainment. To meet the definition of entrainment, a pre-existing internal oscillator should gradually synchronize to an external rhythm[18]. So far, speech tracking has not been proven to fulfill these criteria; it cannot be ruled out that the rhythmic auditory response to speech reflects mere superposition of evoked responses rather than entrainment[21]. Conversely, the music perception literature accumulated more evidence to support synchronization of intrinsic neural oscillations

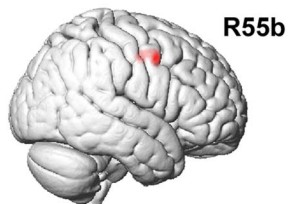

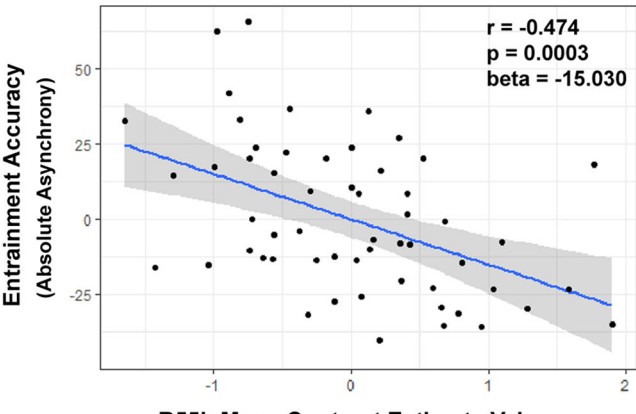

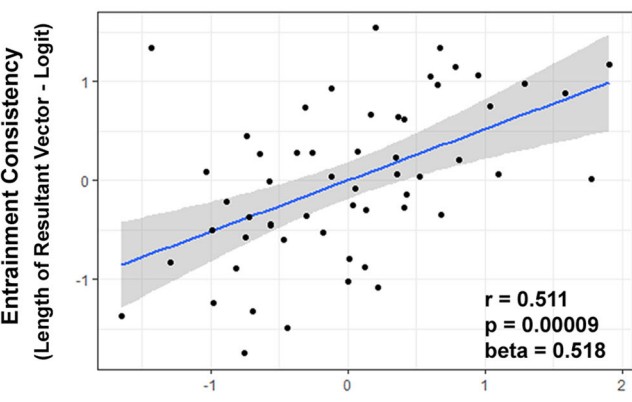

**Fig. 4 Partial regression plots of the relationship between measures of rhythmic entrainment and activity within the R55b.** Significant association was documented between indices of rhythmic entrainment accuracy (absolute asynchrony) and consistency (length of resultant vector–logit) and R55b activity during the Rhythm paradigm (third complexity level vs. baseline), while controlling for age, gender and musical education ($n = 56$). The solid blue line is the regression line; the gray bands represent the standard error of the regression line.

to external rhythms. Several studies have documented neural oscillations at the pulse frequency, even when subjects listened to complex, highly syncopated rhythms, which did not contain the pulse frequency itself[20,47,48]. These studies have also showed induction of activity at harmonics and subharmonics of the beat frequency, which were absent from the stimulus[20,47]. Moreover, mere imagery of metric structures (i.e., march and waltz) was shown to induce subharmonics of the beat frequency[49].

The association of neural oscillations with meter imagery opens another long-standing discussion, of whether or to what extent the automatic, bottom-up process of neural entrainment to the beat is affected by top-down processes[21]. One prevailing opinion poses that neural entrainment is enhanced by selective attention[21,50–52]. The well-supported association between

attention and entrainment is in agreement with the correspondence we found between enjoyment (probably associated with increased attention) and brain (activity)–behavior (rhythmic entrainment) correlation, across different levels of musical complexity. Another view is that motor cortices are involved in top-down control of auditory cortex entrainment to external rhythms[53]. Morillon and his colleagues[54] proposed a mechanism of auditory active sensing, where rhythmic motor activity affects auditory perception through a top-down mechanism involving attention and temporal prediction, a view highly reminiscent of the active inference scheme proposed by Karl Friston[16].

Much of the discussion on neural entrainment to external auditory stimuli has been focused on the auditory cortex; what do we know about entrainment of motor-related regions or the coherence between auditory and motor oscillators in response to auditory stimulus? Modulation of neural oscillations during passive listening to auditory rhythms has been documented outside the auditory region, particularly in motor-related areas, including the precentral gyrus, medial frontal gyrus and the cerebellum[55]. Interestingly, low-frequency oscillations in the premotor region have been reported during exposure to non-musical rhythmic auditory stimulus (shuffled speech)[56]. Similarly, speech tracking (particularly at the phrasal timescale) has been documented in the premotor cortex[57]. As for the interaction between auditory and motor regions, generally speaking, brain-to-brain entrainment has been suggested to underpin communication between different sensory modalities and between the sensory and motor systems[20,50,58]. Moreover, phase synchrony between brain oscillators has been advocated to facilitate information transfer and integration[59,60]. However, while oscillations in the delta (1–4 Hz) and beta (14–30 Hz) bands have been implicated in communication between motor and auditory regions[55,61,62], we are not aware of direct evidence establishing auditory-to-motor neural entrainment at the cortical level.

A related open question is to what extent subcortical, rather than cortical, structures are involved in auditory-motor integration. This question is relevant to an additional interesting result of the current study, namely the relative selectivity of the brain (activity)-behavior (rhythmic entrainment) association to the R55b. A stepwise regression indicated that among the 36 parcels most activated during passive listening to rhythmic patterns, activity in the R55b accounted for much of the variance in the capacity to synchronize to the beat. The importance of this finding is twofold; first, it allows localizing rhythmic entrainment, which has already been linked to the premotor cortex[56,57,63], to a highly restricted area within this strip. Second, it emphasizes the uniqueness of area 55b among cortical regions activated by music. Although much less prominently, the left perisylvian language area (PSL) also showed positive correlation with measures of entrainment consistency. PSL is located at the apex of the posterior Sylvian fissure within the supramarginal gyrus (SMG). Remarkably, Glasser and his colleagues in their seminal work introducing the HCP-MMP1 reported functional connectivity between the PSL and area 55b, more on the left hemisphere, and claimed for its association with a language network[30]. Further, PSL appears to overlap, at least partially, with area Spt (Sylvian-parietal-temporal region)[64], which has previously been linked to auditory-motor integration[10]. Why did other regions fail to emerge, or were even negatively correlated with rhythmic entrainment, as in the case of the frontal operculum 1 (FOP1), and whether this finding implies involvement of subcortical structures in auditory-motor integration, is left for future investigation.

Returning to area 55b, it is important to notice that its location is in line with previous theories of auditory-motor integration and prediction. Located relatively dorsally in the premotor strip[65],

**Table 2 Model summary of the stepwise regression analysis, relation between rhythmic entrainment and activity within multiple brain regions.**

| Dependent variable | Model | R | Adjusted $R^2$ | $R^2$ Change | F Change | Significance F change |
|---|---|---|---|---|---|---|
| AA | (Constant), Age, Gender, MusEdu | 0.120 | −0.042 | 0.014 | 0.253 | 0.859 |
|  | (Constant), Age, Gender, MusEdu, R55b | 0.486 | 0.176 | 0.222 | 14.802 | 0.0003 |
| LRV-logit | (Constant), Age, Gender, MusEdu | 0.252 | 0.009 | 0.063 | 1.174 | 0.328 |
|  | (Constant), Age, Gender, MusEdu, R55b | 0.555 | 0.254 | 0.244 | 18.010 | 0.00009 |
|  | (Constant), Age, Gender, MusEdu, R55b, LPSL | 0.618 | 0.320 | 0.074 | 5.999 | 0.018 |
|  | (Constant), Age, Gender, MusEdu, R55b, LPSL, LFOP1 | 0.667 | 0.376 | 0.062 | 5.493 | 0.023 |
| ENT | (Constant), Age, Gender, MusEdu | 0.237 | 0.002 | 0.056 | 1.029 | 0.387 |
|  | (Constant), Age, Gender, MusEdu, R55b | 0.544 | 0.240 | 0.240 | 17.358 | 0.0001 |
|  | (Constant), Age, Gender, MusEdu, R55b, LPSL | 0.609 | 0.308 | 0.076 | 6.004 | 0.018 |
|  | (Constant), Age, Gender, MusEdu, R55b, LPSL, LFOP1 | 0.668 | 0.378 | 0.075 | 6.637 | 0.013 |

Hierarchical stepwise multiple linear regression ($n = 56$). Dependent variables: absolute asynchrony (AA), length of resultant vector - logit (LRV-logit), entropy of relative-phase distribution (ENT). Independent variables: age, gender, musical education (MusEdu), and mean contrast estimates within the 36 most activated parcels of the Rhythm paradigm for the contrast of the third condition vs. baseline (see Fig. 2, Rhythm, for abbreviated names of the 36 brain regions and Supplementary Data 2 for full names of HCP-MMP1 regions).
*R55b* right 55b, *LPSL* left perisylvian language area, *LFOP1* left frontal operculum 1 region.

**Table 3 Coefficients of stepwise regression models.**

| Dependent variable | Independent variable | Unstandardized coefficients | | Standardized coefficients | *t*-value | *p*-value |
|---|---|---|---|---|---|---|
|  |  | B | Std. error | Beta |  |  |
| AA | Constant | 93.510 | 22.574 |  | 4.142 | 0.0001 |
|  | Age | −0.238 | 0.645 | −0.046 | −0.369 | 0.713 |
|  | Gender | −5.181 | 6.498 | −0.102 | −0.797 | 0.429 |
|  | MusEdu | −1.139 | 1.038 | −0.137 | −1.098 | 0.278 |
|  | R55b | −15.030 | 3.907 | −0.482 | −3.847 | 0.0003 |
| LRV-logit | Constant | −0.107 | 0.709 |  | −0.151 | 0.881 |
|  | Age | −0.009 | 0.019 | −0.056 | −0.500 | 0.619 |
|  | Gender | 0.433 | 0.189 | 0.260 | 2.294 | 0.026 |
|  | MusEdu | 0.058 | 0.030 | 0.212 | 1.935 | 0.059 |
|  | R55b | 0.513 | 0.113 | 0.501 | 4.536 | 0.00004 |
|  | LPSL | 0.227 | 0.074 | 0.350 | 3.060 | 0.004 |
|  | LFOP1 | −0.263 | 0.112 | −0.262 | −2.344 | 0.023 |
| ENT | Constant | 0.077 | 0.063 |  | 1.220 | 0.228 |
|  | Age | −0.00005 | 0.002 | −0.004 | −0.033 | 0.974 |
|  | Gender | 0.035 | 0.017 | 0.233 | 2.063 | 0.044 |
|  | MusEdu | 0.006 | 0.003 | 0.251 | 2.290 | 0.026 |
|  | R55b | 0.045 | 0.010 | 0.494 | 4.472 | 0.00005 |
|  | LPSL | 0.021 | 0.007 | 0.359 | 3.144 | 0.003 |
|  | LFOP1 | −0.026 | 0.010 | −0.287 | −2.576 | 0.013 |

For details, see Table 2.

area 55b may belong to the dorsal auditory stream, which has been advocated to mediate auditory-motor integration and prediction by Rauschecker[12] and others[66], and was also implicated in rhythmic entrainment within the ASAP thesis[5]. Interestingly, the Spt region (which partly overlap with area PSL) has also been linked to the dorsal stream of auditory processing and language[10,66]. The Habitual Pragmatic Event Map framework by Ricarda Schubotz posits that precompiled action templates stored in the premotor cortex are central to prediction of perceptual (including musical) events, and that the exact portion of the premotor strip involved depends on the type of the related action[13]. The exact localization for rhythmic events has not been fully characterized. In any case, the premotor cortex has been implicated by both Rauschecker[67] and Schubotz[68] in predicting the structure of acoustic sequences, which are the backbone of music and speech/language.

As the mirror neuron system has also been linked to sensorimotor association[69,70], a question is raised whether area 55b belongs to the mirror neuron network (for review on mirror neurons see Rizzolatti et al.[71]). Albeit first identified in the ventral premotor cortex of the macaque monkey (area F5)[71], mirror neurons have later been documented also in the dorsal premotor region (area F2vr and F7) and other cortices (parietal and temporal) of the macaque brain[72–74]. Neuroanatomical localization of mirror neurons in the human brain is still an area of controversy, nevertheless, both ventral and dorsal premotor cortices have been proposed as central nodes of the human mirror neuron network[75–77]. Support for mirror properties of area 55b can be found in fMRI studies of sensorimotor synchronization reporting coordinates similar to those of area 55b (bilaterally) while synchronizing via finger tapping to the musical beat[9,78–83]. Notably, Chen et al.[9] have directed attention

to the border between the dorsal and ventral premotor cortices (mid-premotor cortex), which emerged as a core region of both passive perception and sensorimotor synchronization. The reported coordinates of the mid-premotor cortex are highly concordant with area 55b.

Due to the emerging relevance of area 55b to music and language, it is intriguing to explore whether it is more specifically involved in an auditory-vocal mirror system, perhaps even represent a human homolog of the avian high vocal center, the highest nucleus among birdsong nuclei identified so far, which has been proposed to mediate complex vocal learning through mirroring, sequencing and rhythmic bursting[84,85], and has been related to the mammalian premotor cortex[84]. Though audio-vocal mirroring in humans has been linked traditionally to Broca's area[84], as far as we can tell, regions at the vicinity of area 55b appear to have emerged in previous studies directed at activating an auditory mirror circuit[86,87]. Other studies have associated nearby locations with a pre(motor) vocal network[6,88]. Unfortunately, knowledge about the cortical control of the complex neuromuscular apparatus coordinating human laryngeal vocalizations is limited. At least two distinct loci have been reported within the pre(motor) region, the ventral and the dorsal laryngeal motor cortices[88–90]. Albeit plausible, whether the dorsal laryngeal motor cortex matches area 55b is still to be determined. If confirmed it will provide an anatomical proof for the presumptive linkage between vocal functioning and rhythmic entrainment and may reconcile the old controversy of the origin of music, vocalization vs. rhythmic percussion. Here it is also interesting to mention, that the Spt, probably a part of the PSL, which emerged in our analysis as weakly associated with measures of rhythmic entrainment, has been advocated to take part in a sensorimotor integration circuit dedicated for the vocal tract[66]. The current study has focused on the role of the R55b in music perception, to what extent this region is also involved in other brain processes, including music/speech production, speech perception, and visual perception of music and language remains to be determined by further research.

To conclude, building on the current observations and prior theories, area 55b is advised as an epicenter for perception-action coupling within a nexus of auditory-motor loops mediating sensorimotor integration and implicit prediction[3,91,92]. We propose that area 55b in humans may tie up advanced mechanisms of hierarchical sensorimotor integration via neural entrainment with mechanisms of complex vocal control, thus enabling the peculiar evolution of music and speech, which differentiates us from non-human primates, on the one hand, and from avian vocal learners, on the other hand. Future definition of area 55b's connections to other brain regions and the functions these connections harbor would probably contribute to better understanding of sensorimotor integration, a fundamental brain process which on the one side of the spectrum, characterizes most elementary behaviors, such as goal-directed movement of even the simplest organism, and on the other side, set the stage for human unique complex behaviors, probably representing a late evolutionary form of hierarchical cognition. Beyond theoretical interest, improved understanding of these cardinal processes and their anatomical infrastructure will likely promote interventions for relevant clinical conditions, such as movement disorders (e.g., Parkinson's disease), developmental learning disabilities (e.g., developmental dyslexia), and social cognition impairment (e.g., autism spectrum disorders). Furthermore, it may accelerate human brain-inspired approaches in artificial intelligence and robotics, which so far have been advanced through reciprocal interactions with the neuroscientific understanding of implicit learning and prediction[93].

## Methods

**Participants.** In total, 71 right-handed healthy volunteers participated in the study (median age, 25.0 years (range, 18–44); female, 41; median general education, 14 years (range, 12–18 years), median musical education, 2.0 years (range, 0–12 years)). Fifty-nine out of the 71 subjects (median age, 25.0 years (range, 19–42); female, 35; median general education, 14 years (range, 12–18 years), median musical education, 2.0 years (range, 0–12 years)) participated in both the behavioral and imaging sessions. All subjects had normal or corrected-to-normal vision, reported normal hearing, had no history of neurological or psychiatric disorder, no history of substance/alcohol abuse, and no structural brain abnormality. All subjects were eligible for MRI scanning and did not use medications that may interfere with the study. Applicants with professional background in music or dance were excluded. Past musical experience was determined through a musical experience questionnaire. To ensure normal music perception, a brief version of the Profile of Music Perception Skills (PROMS, https://www.uibk.ac.at/psychologie/fachbereiche/pdd/personality_assessment/proms/) was administered. The study was conducted according to the guidelines of the Declaration of Helsinki and approved by the Institutional Review Board of Tel Aviv Sourasky Medical Center (committee reference number 0017-18-TLV). Informed consent was obtained from all subjects involved in the study.

### Stimuli

*Behavioral task.* Rhythmic entrainment skills were evaluated outside the magnet through a sensorimotor synchronization task adopted from Fujii and Schlaug[38]. The music tapping test of the Harvard Beat Assessment Test[38] is composed of three musical excerpts from the Beat Alignment Test[94]: Hurts So Good by J. Mellencamp (rock style, duration = 14 s), Tuxedo Junction by Glenn Miller (jazz style, duration = 16 s), and A Chorus Line by Boston Pops (pop-orchestral style, duration = 14 s). Each tune was presented at three different tempi, 100, 120, and 140 beats per minute. A 1000 Hz pure tone of 200 ms duration preceded each musical excerpt to signal the beginning of the trial. Materials were downloaded from https://www.ncbi.nlm.nih.gov/pmc/articles/PMC3840802/ (GUID: D30B541A-F157-4B50-AFB2-ECBFD18500D1).

*Rhythm fMRI paradigm.* Out of 50 drum-breaks introduced by Witek et al.[95], 14 excerpts were chosen to fit one of four levels of rhythmic complexity in the form of syncopation: (I) isochronous rhythm, (II) mildly syncopated, (III) moderately syncopated or (IV) highly syncopated rhythm. Two isochronous drum-breaks were composed by the authors to conform to level I. Each drum-break consisted of a two-bar phrase in 4/4 time looped four times at 110 beats per minute to yield a 17.45 s excerpt. Syncopation level was determined by two indices adopted from the literature: (1) The C-score offered by Povel and Essens[96], and (2) A syncopation index suggested by Fitch and Rosenfeld[97]. Excerpts were generated using Studio drummer of the Native Instruments Komplete, in Cubase Pro 9.5 (Steinberg Media Technologies GmbH). The paradigm consisted of four conditions (four levels of rhythmic complexity), four blocks per condition, block duration 18 s, inter-block interval 9 s. The first 30 s (including an additional 9 s drum-break) were discarded from analysis. Total task time was 8 min and 9 s (Supplementary Fig. 1).

*Melody fMRI paradigm.* Eight 15 s excerpts were extracted from Mozart quartets available in KernScores (http://humdrum.ccarh.org/). Excerpts were drawn from the highest melodic line (mostly first violin, flute/oboe one excerpt each) to fit into one of two melodic complexity levels, based on pitch unexpectedness (information content) determined by the algorithm IDyOM (Information Dynamics of Music, http://mtpearce.github.io/idyom/[98,99]). Other musical features, such as pitch range, mode, meter, tempo, density, and rhythm complexity were kept as invariable as possible. The IDyOM model predicted unexpectedness of subsequent notes based on chromatic pitch, chromatic pitch interval, chromatic interval from tonic and contour, using both short-term and long-term models. The long-term model was trained on nine datasets of western music (Bach chorales and several folk song cohorts available in KernScores). Eight additional melodies were generated using the above Mozart's excerpts: four by reducing the lower complexity excerpts into single pitch (tonic) tunes, and additional four, by randomizing notes of the higher complexity excerpts, using the shuffle pitches function in Sibelius software (Version 7.5). All excerpts were recorded with piano timbre (The Grandeur, Native Instruments Komplete) using Cubase Pro 9.5 (Steinberg Media Technologies GmbH). The paradigm consisted of four conditions (four levels of melodic complexity), four blocks per condition, block duration 15 s, inter-block interval 9 s. The first 36 s (including an additional 15 s melody) were discarded from analysis. Total task time was 7 min and 27 s (Supplementary Fig. 2).

*Harmony fMRI paradigm.* Eight-chord descending-fifths sequences were constructed following classical paradigms of musical prediction. Progressions were ended on either the tonic chord (regular cadence), the sixth-degree chord (less regular (deceptive) cadence) or the Neapolitan chord (irregular cadence). The Neapolitan chord is a major chord built on the lowered 2nd scale degree, not a dissonance by itself, but as a cadence sounds completely unexpected[100]. A scrambled version of the chord progression ending on the tonic was also included.

Duration of chords 1–7 in each sequence was 600 ms, while the final chord lasted 1200 ms. Sequences were recorded with piano timbre (The Grandeur, Native Instruments Komplete) using Cubase Pro 9.5 (Steinberg Media Technologies GmbH). Each sequence was transposed into 12 major keys. The paradigm consisted of four conditions (Tonic, Sixth degree, Neapolitan, Scrambled), four blocks per condition, block duration 18 s, inter-block interval 9 s. The first 30 s (including a 12 s block of chord sequences) were discarded from analysis. Total task time was 8 min and 9 s (Supplementary Fig. 3).

**Procedure.** The combined behavioral and neuroimaging study was conducted at Sagol Brain Institute, Tel Aviv Sourasky Medical Center. On the first session, participants completed a battery of cognitive tasks outside the magnet, including a sensorimotor synchronization task (the music tapping test) drawn from the Harvard Beat Assessment Test[38]. Stimulus presentation and response recording were controlled by Presentation software version 20.1 (Neurobehavioral Systems, Albany, USA). Auditory stimuli were delivered over headphones (Sony MDR-7506). Participants were instructed to tap the quarter-note beat underlying the musical excerpts (demonstration by the experimenter and a few practice trials preceded actual task performance). The nine musical excerpts were repeated twice for each participant (the order of stimuli presentation was randomized). Exact timings of finger taps were captured by the TapArduino device[101]. Participants used the index finger of their dominant hand to tap on a force-sensitive resistor pad connected to the Arduino. No auditory feedback was provided during tapping. Tap timings registered by Presentation software were subjected to further analysis as detailed below. On the second session, participants underwent fMRI scanning. They were instructed to lie as still as possible (refrain from any movement) and listen carefully to the presented auditory stimuli while fixating on a central cross (projected on a screen at the back of the bore and viewed through a mirror attached to the top of the head coil). They were not informed about the nature of the musical stimuli. Stimuli were presented in a block design using Presentation software version 20.1 (Neurobehavioral Systems, Albany, USA). At the end of the scanning session, subjects listened again to all musical stimuli and reported their emotional response to each musical block through five-point Likert scales. In the majority of cases, participants completed all parts of the study within one week. In any case, time delay between sessions did not exceed 5 weeks. Sixteen out of the 59 (27%) subjects who participated in both sessions, underwent the fMRI scanning before the behavioral session.

**fMRI acquisition.** Scanning was performed on a 3.0 Tesla MAGNETOM Prisma MRI scanner (Siemens, Munich, Germany) using 20-channel head coil. The scanning session included a Magnetization Prepared RApid Gradient Echo (MP-RAGE) sequence to provide high-resolution structural images (TR/TE = 1860/2.74 ms, flip angle = 8°, FOV = 256 × 256 mm, voxel size = 1 × 1 × 1 mm, 176 slices). Functional scans were acquired using a T2*-weighted echo planar-imaging (EPI) sequence (TR/TE = 3000/35 ms, flip angle = 90°, 96 × 96 matrix, FOV = 220 × 220 mm, 46 slices of 3 mm thickness, no gap, whole-brain coverage).

**fMRI analysis.** Preprocessing of functional images was performed using fMRIPrep 20.0.2[102]. Briefly, functional images were corrected for slice timing and distortion, realigned, co-registered with the structural image, normalized into MNI space (MNI152NLin2009cAsym), and smoothed with a 6 mm full width half-maximum Gaussian kernel. Participants exhibiting head motion of >2 mm were excluded from analysis of the relevant task, i.e., four, five, and six participants in the rhythm, melody, and harmony paradigms, respectively. First and second-level analyses were conducted using the Statistical Parametric Mapping (SPM12) software package (http://www.fil.ion.ucl.ac.uk/spm) implemented in MATLAB (version R2018a, Mathworks, Natick, USA). The blood oxygenation level-dependent (BOLD) fMRI signal was modeled with a general linear model, using a canonical hemodynamic response function and a standard temporal filter of 128 s. The following confound regressors, computed by fMRIPrep, were included in the analysis: time series derived from the whole brain, white matter and cerebrospinal fluid masks, six rigid-body motion parameters, the first temporal derivatives and quadratic terms for each of the above, framewise displacement and the standardized derivative of root mean square variance over voxels. For motion scrubbing, framewise displacement threshold of 0.9 mm was applied.

For the whole brain analysis, individual statistical parametric maps were calculated for the basic contrast of all conditions together vs. baseline (no auditory stimulus) to yield maximal activations. Using Python version 3.7.9 (https://www.python.org/), including NumPy[103] and SciPy[104], group-level statistical parametric maps (family-wise error (FWE) corrected, p < 0.05), were overlaid on a projection of the HCP-MMP1 parcellation[30] onto the ICBM 152 2009c non-linear asymmetric version of the MNI space, downloaded from NeuroVault (https://neurovault.org/images/29489/). Mean intensity level per parcel was calculated for each activation map and the first 36 most activated regions (top ten percent of all parcels) were highlighted on a parcellation map. 3D Slicer software (version 4.11, Kitware, Inc., New York, NY, and Brigham and Women's Hospital, Boston, MA) was used to visualize the volumetric parcellations. Figures were assembled using Adobe Photoshop version 10.0.

For the region of interest analysis, using SPM12, individual mean contrast estimates were extracted from the R55b parcel, for each condition (vs. baseline) of each paradigm. In addition, mean contrast estimate values were extracted from each of the 36 most activated parcels of the Rhythm paradigm, for the third condition (moderately syncopated sequences). As the conversion of the HCP-MMP1 parcellation from surface- to volume-based coordination system has been criticized[105], mean contrast estimates were also extracted from a 6 mm sphere region-of-interest centered on the extra-auditory peak cerebral activation, to replace contrast estimates derived from the R55b parcel in a repeated analysis.

**NeuroSynth meta-analysis.** We used NeuroSynth, a web-based platform for large-scale automated synthesis of fMRI data (https://neurosynth.org/[37]), to produce a meta-analytic activation map of 163 imaging studies using the term music. The Neurosynth term-based meta-analysis tool applies text-mining techniques to identify neuroimaging studies that used specific terms of interest (hundreds of psychological concepts). Published articles are automatically parsed and each article is tagged with a set of terms, which occur at least once in their abstract. Activation coordinates are also automatically extracted from each article. For each term of interest, an association test map is constructed to display brain regions that show statistically significant association with the term. For instance, a positive voxel in the association test map for music indicates that studies tagged with the term music are more likely to report activation at that voxel relative to other studies. The statistical computation is based on chi-square test of 2 × 2 contingency table with the factors (1) term inclusion (mentioned or not mentioned in a study's abstract) and (2) voxel activation (reported or not reported in a study). Voxel Z-scores correspond to p-values of the chi-square test. Maps are corrected for multiple comparisons using a False Discovery Rate (FDR) criterion of 0.01 (for details, see Neurosynth.org/faq).

At the time of our analysis, the NeuroSynth database included activation coordinates from >14,000 neuroimaging studies. Details of the 163 studies used for the meta-analysis of the term music can be found in https://neurosynth.org/analyses/terms/music/ (see Supplementary Data 1). To overlay the NeuroSynth map on the HCP-MMP1 parcellation, we here first projected the parcellation onto the ICBM 152 non-linear 6th generation version of the MNI space, using a MATLAB code based on: https://figshare.com/articles/dataset/HCP-MMP1_0_projected_on_MNI2009a_GM_volumetric_in_NIfTI_format/3501911/4. Gray matter segmentation was generated in SPM12 using a T1 brain template included with FSL (Functional Magnetic Resonance Imaging of the brain (FMRIB) Software Library, Oxford, UK). 3D Slicer software (version 4.11, Kitware, Inc., New York, NY, and Brigham and Women's Hospital, Boston, MA) was used for visualization. Figures were assembled using Adobe Photoshop version 10.0.

**Behavioral task analysis.** Using MATLAB (version R2020b, Mathworks, Natick, MA), timings of taps performed during the 18 musical excerpts of the music tapping test were extracted from the Presentation logfile for each participant. Musical beat timings were detected using BeatRoot 0.5.8 (https://code.soundsoftware.ac.uk/projects/beatroot) with minor manual adjustments. To assess the degree of synchronization between participant's taps and the beat we used both linear and circular methods[38,39]. The first five and last three taps of each trial were discarded from further analysis. First, tap timings were aligned to the stimulus beat timings on a linear scale. For the assessment of synchronization accuracy, absolute asynchrony (AA) was calculated as the mean of absolute (not signed) distances in milliseconds between taps and corresponding beats across trials[39]. Lower AA scores indicate higher accuracy. For the assessment of synchronization consistency, two circular statistics indices were computed as recommended by Fujii and Schlaug[38]: (1) length of resultant vector (LRV) and (2) entropy of relative-phase distribution (ENT). Time differences between taps and corresponding beats were transformed into angles measured in radians, considering the inter-stimulus interval of the relevant trial as the circle circumference. LRV, representing the distribution of the relative phase angles, was calculated using circ_r function of the CircStat toolbox[106] implemented in MATLAB. LRV ranges from 0 to 1, where 1 denotes perfect concordance of angles. Following Dalla Bella et al.[39], LRV values were submitted to logit transformation, to reduce data skewness. ENT also represents circular spread and ranges from 0 to 1, yet, it uses Shannon entropy[107], i.e., the logarithms of the probability density function, to allow differentiation of random phase distribution from the mixture of in-phase and anti-phase locking[38].

**Statistics and reproducibility.** The overall sample size (N = 71) was determined based on literature recommendations[108,109] and prior lab experience with similar study designs. The sample size differed for each analysis, as not all participants completed all tasks and since exclusion from the fMRI analysis was based on excessive head motion during a specific scan. The exact composition of subjects is therefore not equal for all tasks but contain minor differences. We preferred differential removal of data according to its quality to keeping the exact same subjects in all paradigms, in order to achieve maximal statistical power for each analysis. Final sample sizes were as follows: Rhythm paradigm, fMRI only, n = 67; Melody paradigm, fMRI only, n = 66; Harmony paradigm, fMRI only, n = 65; Rhythm paradigm, combined study, n = 56; Melody paradigm, combined study, n = 58;

Harmony paradigm, combined study, $n = 53$; Emotional ratings for all fMRI paradigms, $n = 62$.

Computation and graphical presentation of correlations between behavioral (synchronization accuracy (AA) and consistency (LRV-logit and ENT)) and imaging (mean contrast estimate values within the R55b) measures were conducted using GraphPad Prism version 9.4.0 for Windows (GraphPad Software, California, USA). Spearman's correlation was chosen since part of the variables were not normally distributed.

Linear mixed model analysis of the relationship between pleasure ratings and experimental condition was performed using R software version 4.0.4 (R Foundation for Statistical Computing, Vienna, Austria). The analysis was conducted for each paradigm separately using the functions lme() and anova () (package nlme), with experimental condition as fixed effect and subject and block as random effects. Boxplots were generated using ggplot() (package ggplot2). Post hoc comparisons between conditions of each paradigm were calculated using emmeans() (package emmeans), with Tukey's correction for multiple comparisons. Parameters of descriptive statistics were calculated using tapply() (package base).

R software version 4.0.4 (R Foundation for Statistical Computing, Vienna, Austria) was used to perform multiple linear regression analyses to evaluate the association between activity within the R55b (independent variable) and the rhythmic entrainment indices (dependent variables), while controlling for age, gender and musical education. This analysis was conducted separately for each of the metrics of rhythmic entrainment, i.e., AA, LRV-logit and ENT, and each of the conditions of interest, i.e., third, second, and first conditions of the Rhythm, Melody, and Harmony paradigms, respectively (the conditions characterized by maximal correlation between R55b activity and entrainment indices). The analysis was performed using the R lm() function (package stats). Numerical results were attained through the summary() function. Partial regression plots were generated using the ggplot() function (package ggplot2).

Finally, a hierarchical stepwise linear regression was performed using the Statistical Package for the Social Sciences (IBM SPSS statistics 20.0). This analysis was conducted separately for each of the measures of rhythmic entrainment, which served as dependent variables. Mean contrast estimate values extracted from the 36 brain parcels most strongly activated during the Rhythm paradigm served as independent variables of interest. Age, gender, and musical education were forced-entered in the first block of the hierarchical regression to ensure that the contribution of localized brain activity is evaluated while these covariates are controlled for. The 36 variables of localized brain activity were included in a second block, which was analyzed using the bidirectional stepwise selection method. At each step, variables were chosen based on $p$-values: entry criterion, $p < 0.05$, removal criterion, $p > 0.1$. Stepwise regression was chosen in this case due to the large number of independent variables and the expected multicollinearity among the brain activity variables.

**Reporting summary**. Further information on experimental design is available in the Nature Research Reporting Summary linked to this paper.

## Data availability

Musical stimuli which have been used in the three fMRI paradigms as well as the statistical parametric maps underlying Fig. 2 are available to download from the Open Science Framework (OSF). https://osf.io/qnbwv/?view_only=e67e4737c321450f8488a80dcbfd5f5d. Behavioral and imaging raw data that support the findings of this study are available from the corresponding author upon reasonable request.

## Code availability

The custom Python (version 3.7.9) script used to overlay statistical parametric maps onto the volumetric version of the HCP-MMP1 parcellation and estimate brain activity per parcel is available to download from the Open Science Framework (OSF). https://osf.io/qnbwv/?view_only=e67e4737c321450f8488a80dcbfd5f5d (https://doi.org/10.17605/OSF.IO/QNBWV).

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

## Acknowledgements

This work was supported by the European Union's Horizon 2020 Framework Programme for Research and Innovation under Specific Grant Agreement No. 945539 (Human Brain Project SGA3) and the Sagol Family Fund. We would like to thank Prof. Yaniv Kanat-Maymon and David Harar for statistical consultation, Adi Sarig, Karni Bar-Or, Guy Gurevitch, Dr. Yulia Lerner, Dr. Neomi Singer, Itamar Jalon, Avihay Cohen, Ayam Greental, Oren Levin, Yael Hamrani, Dr. Moran Artzi, and Prof. Dafna Ben-Bashat for helpful discussions and/or technical assistance with experimental design and data analysis. We also thank Prof. Zohar Eitan and Prof. Matitiahu Mintz for insightful discussions.

## Author contributions

T.S., C.R.G., R.Y.G., and T.H. contributed to the conception and design of this work. T.S., N.A., O.Shany, O.Shuster, A.L., and R.Y.G. were involved in experiment preparation and data collection. T.S., N.A., and O.Shany carried out data analysis. T.S., O.Shany, C.R.G., R.Y.G., and T.H. contributed to interpretation of the results. T.S. drafted the manuscript, which was revised by C.R.G., R.Y.G., and T.H. All authors read and approved the final version of the manuscript.

## Competing interests

The authors declare no competing interests.
