## [Peer Review File · Communications Biology]

Reviewers' comments:

Reviewer #1 (Remarks to the Author):

The current study we used a meta-analysis of music imaging studies; and three in-house fMRI paradigms of rhythm, melody and harmony perception were applied. The results of an accompanying behavioral study (complementary to the fMRI data) were subjected to inter-individual brain-behavior analysis.

The authors discovered that there was a significant effect of the mean activity within the R55b during the rhythm experiments, on accuracy and consistency indices of rhythmic entrainment. But nothing noteworthy was found in neighboring RFEF. No significant results were obtained for the left 55b or FEF. A similar analysis of R55b activity during the melody and harmony experiments did not reach significance.

The paper makes an interesting addition to the literature on music perception and furthers our understanding of motor networks in beat processing.

I have a few comments to improve the paper.

- 1) Please provide a PRISMA flowchart for the fMRI meta analysis. The keyword "music" seems to capture a lot of the literature. More detail needs to be provided regarding this procedure for replicability. There are several other ALE procedures that have been done on similar data, it would be important to specify the methodology here to compare the present data to previous studies.
- 2) Borderline effects are neither here nor there. Statistical significance (for better or worse) is binary. Results approaching significance have to be reported as such. It would be useful for the authors to clarify this, since the melody/harmony results are quite clear.
- 3) The discussion on the mirror neuron system is a bit weak. I would instead encourage the authors to not speculate here and instead stick with the well understood dorsal stream mechanisms.

Overall, this is a very interesting paper. With appropriate revisions, it could have a significant impact on the field of the biology of music perception in humans (and other animals).

Reviewer #2 (Remarks to the Author):

Review for "The Rediscovered Motor-related Area 55b Emerges as a Core Hub of Music Processing"

General impression

This study provides useful insights on the specific role of R55b in music perception, and possibly generalizing to other functions, such as speech perception. The methods seem solid and rich, including both a meta-analysis and a targeted fMRI study. I particularly enjoyed the choice of naturalistic paradigm, in which participants listen to the stimuli without explicit task to focus on specific aspects of the stimuli. If the following comments are taken into account, mainly regarding further clarifications and connections to other functions, I would recommend acceptance for publication, as this study would contribute very valuable knowledge to the field.

Specific comments

Lines 74-75: "thus might represent the right homologue of the language hub advocated lately" this phrasing softly implies a left-right distinction between language and music. We indeed know that language is heavily left-lateralized, but especially speech comprehension is also distributed to both hemispheres and shares networks with other functions, such as auditory perception. So I would recommend rephrasing this sentence to not imply this binary hemispheric distinction, but to leave open the possibility that the right 55b might be also used in speech comprehension, as part of the

auditory processing network (of course used for music as your study is focusing on).

Line 77: "probably", is it not clear how many of the studies were testing the production, and how many studies tested the perception of music? Please provide the details, possibly in percentage of the total studies entering the meta-analysis.

Lines 87-88: "location 24, 32 and 29 in the rhythm, melody and harmony paradigms, respectively," this is not clear as the format of the paper for this journal presents the results before the design. Please describe with words what were the designated locations, so that it is clear when one reads the paper serially.

Lines 91-111: This paragraph is not completely consistent, it starts with referring to motor regions, then touches upon the concepts of entrainment and synchronization, and quickly moves on to beat perception via prediction. I would recommend making it more coherent by omitting some of the details, and keeping only the necessary information for the point you need to make, as for example the phenomenon of negative asynchrony. Also please use a topic sentence (the first sentence of the paragraph) that states the argument you want to make in this paragraph and paves the way for the details provided next in the paragraph.

Line 113: "rhythmic entrainment" please clarify whether you mean endogenous entrainment of the neural circuits or synchronization to beat as caused by external stimulus, or what exactly is the process you are assuming that your results track. (see for example: <https://journals.sagepub.com/doi/full/10.1177/0748730420972817> or a bit more relevant to the neural circuits: <https://www.frontiersin.org/articles/10.3389/fnhum.2018.00263/full> and [https://www.cell.com/trends/cognitive-sciences/fulltext/S1364-6613\(20\)30078-4](https://www.cell.com/trends/cognitive-sciences/fulltext/S1364-6613(20)30078-4)) There has been confusion in the field in the recent 2-3 years and new distinctions have emerged, also in the field of language processing, so it would be very useful to clarify with regard to music.

Line 114: "59 subjects also performed a sensorimotor synchronization task outside the scanner", did this happen before or after the fMRI measurements? If before, you need to acknowledge that there might be effects of this task on the passive task inside the scanner. I see now that this is in a separate session, maybe add a clarification in the sentence, stating that the fMRI measurements were done during the second session.

Lines 120-121: "the rhythm paradigm" this needs to be clarified, as the reader who reads linearly won't know what the rhythm paradigm consists of. I would recommend to substitute this phrase with a short description of what the paradigm asked of the people, e.g. "passively following the rhythm of the music piece" or some similar phrase that describes the function of the paradigm, rather than simply naming it.

Lines 132-133: "area 55b is proposed as an epicenter for perception-action coupling within a nexus of auditory-motor loops mediating sensorimotor integration and implicit prediction", I like this definition, thank you. As you are being very specific (and well done) in this sentence, I would recommend adding one more sentence, possibly connecting to the more general functions of music perception and production, as well as speech perception and production. Would be nice to know what you propose with regard to these higher level functions, based on the specificity of this region. I see that you have a very interesting paragraph starting in line 144, but it would still be nice to add one sentence connecting to higher order function, else I'm left wondering about the very specific role without being able to generalize it to the functions I'm interested in.

Figure 1, panels A and B: It is clear from the coordinates at the figure caption that the yellow crosshair is in the right hemisphere, but I would also recommend adding the R and L letters on the corresponding sides of the template in panels A and B, adding visual clarity and making the figure able to almost stand alone.

Figure 2, lines 235-236 "to either rhythmic passages (N=67), instrumental melodies (N=66) or harmonic progressions (N=65)" it is not clear why there are different N in each contrast, didn't all participants do all 3 paradigms? As you are referring to one study, I am assuming that all 71 participants were tested in all 3 paradigms in a block design? Please clarify if that is indeed the case and the length of each block in time. And then also please clarify the small deviations, why some participant's data was discarded per paradigm, and was it the same participant for all three paradigms or not?

Line 297 "Participants": You do not state the ethical approval for this study. This is very important information and needs to be added in the Participants section (unless I have missed something about Nature Communications regulations). If no ethical approval was obtained for this study, I would not recommend publication.

Lines 395-7, "Participants exhibiting head motion of >2 mm were excluded from analysis of the relevant task (four, five and six participants in the rhythmic, melodic and harmonic paradigms, respectively)." Please clarify why you excluded participants based on paradigm, instead of the whole participant, because this creates slightly different groups per condition and it is not clear why this is needed. I understand you do not compare stats between paradigms, which is good, given that you excluded datasets based on paradigm, but it would still be good to be transparent about how much overlap the different groups had, after exclusion of certain datasets per paradigm.

Lines 421-4 "For each paradigm we used the contrast between the condition of maximal enjoyment as documented by post-scan reports (rhythm: third level of complexity, melody: second level of complexity, harmony: regular cadence) and baseline, as maximal enjoyment is expected to characterize periods of maximal entrainment" This is not clear, what is the "condition of maximal entrainment", how was this "documented".

Line 474 "accuracy (AA) ": it is not clear to me why absolute asynchrony (AA) is a measurement of accuracy, please clarify this connection.

Lines 472-9 "Statistical analysis" Please provide a detailed description of the models you calculated and whether you compared different models or you just looked into the estimates of the theoretically hypothesized model. It is not clear which functions of R you used, e.g. lmer(), anova(), summary() (?), and what the p values on the tables correspond to, effects inside each model, or differences between model fit?

Reviewer #3 (Remarks to the Author):

Authors presented a manuscript on the pre-motor area 55b, suggesting it as a core hub of music processing. To do so, they computed a NeuroSynth meta-analysis, three original fMRI tasks and one behavioral study related to music perception.

Although the work may be of interest, the manuscript does not look adequate for the standard of Communications Biology. Indeed, introduction, results and discussion are mixed in a very short text describing previous literature and the current studies. Moreover, no statistical results are reported in the main text. They are shown only in Figures and Tables. I am not necessarily against this procedure, but it seems very far from the normal guidelines provided by Communications Biology. Thus, I would encourage authors to provide a more organized manuscript with a broader literature review and discussion of the results and a better presentation of their results. As follows, I report some guidelines offered by Communications Biology: <https://www.nature.com/documents/commsj-life-style-formatting-guide-accept.pdf>

If the editor is willing to allow authors to perform an in-depth revision of the manuscript based on the above points, here there are a few more technical comments that authors should address.

-Participants: it would be useful to have complete demographic information for the subsample of participants who took part in the behavioral study only. Moreover, the name of the "musical experience questionnaire" that authors used should be reported.

-fMRI tasks: I think authors did one fMRI study with three tasks and not three different fMRI tasks. Moreover, authors should better clarify what they mean by "music processing" since throughout the manuscript they talk about passive music listening, music perception, music entrainment, appreciation of music, etc. with regards to their analyses and choices. Along this line, authors used only the maximal enjoyed musical condition when computing the ROI analysis, claiming that this is connected to entrainment. Here, I would ask authors to show the results for all conditions (i.e. also for the ones where the music appreciation was low). Moreover, it seems that authors centered their manuscript on the concept of music perception/processing, while for the ROI analysis they are specifically talking about music entrainment or enjoyment of music, which is not the same thing.

-NeuroSynth meta-analysis: authors must report more details on how they did the screening in the review phase. Apparently, they started from more than 14000 papers using only the word "music". This does not sound like a good search, and it is not clear how they ended up having 163 studies. I would advise to do a systematic review following Prospero's guidelines, and, even if they still want to do a non-systematic review, authors should at least provide more details on their procedure. Moreover, no details are present about the statistics used in the meta-analysis. This should be reported as well.

Ref: COMMSBIO-21-3651-T

Title: The Rediscovered Motor-related Area 55b Emerges as a Core Hub of Music Perception

Authors: Tali Siman-Tov, Carlos R. Gordon, Netanel Avisdris, Ofir Shany, Avigail Lerner, Omer Shuster, Talma Hendler, Roni Y. Granot

Response to reviewers

We greatly appreciate the reviewers' careful and supportive assessment of our manuscript, the insightful comments and the detailed recommendations, which have helped us strengthen the paper. The resubmitted manuscript has been extended and substantially revised to meet all recommendations as well as the journal guidelines. Below please find a point-by-point reply to the comments.

Reviewer #1

The current study we used a meta-analysis of music imaging studies; and three in-house fMRI paradigms of rhythm, melody and harmony perception were applied. The results of an accompanying behavioral study (complementary to the fMRI data) were subjected to inter-individual brain-behavior analysis.

The authors discovered that there was a significant effect of the mean activity within the R55b during the rhythm experiments, on accuracy and consistency indices of rhythmic entrainment. But nothing noteworthy was found in neighboring RFEF. No significant results were obtained for the left 55b or FEF. A similar analysis of R55b activity during the melody and harmony experiments did not reach significance.

The paper makes an interesting addition to the literature on music perception and furthers our understanding of motor networks in beat processing.

We thank the reviewer for acknowledging the importance of this study.

I have a few comments to improve the paper.

1) Please provide a PRISMA flowchart for the fMRI meta analysis. The keyword "music" seems to capture a lot of the literature. More detail needs to be provided regarding this procedure for replicability. There are several other ALE procedures that have been done on similar data, it would be important to specify the methodology here to compare the present data to previous studies.

We thank the reviewer for this important comment. In the current study, we used the NeuroSynth online database to derive meta-analysis of neuroimaging studies related to music. The purpose of this part of the study was to show that area 55b emerges as a hub of music processing, even when applying a coarse method of literature capturing. The meta-analytic map was downloaded from the NeuroSynth website and then overlaid on the HCP-MMP1 parcellation. As we did not perform the screening and selection phases of the analysis, we

cannot provide a PRISMA flowchart. Yet, in the current version of the manuscript, the Methods section describing the NeuroSynth meta-analysis has been expanded to include information about the meta-analytic methods used by NeuroSynth (page 21, lines 685-699). Details of the 163 studies included in the meta-analysis are now provided in Supplementary Table 1, and a summary of the relevant details is provided in the Results section (page 4, lines 139-143). Further, the second paragraph of the Discussion now refers to similar previous meta-analyses of music perception (page 7, lines 269-273).

2) Borderline effects are neither here nor there. Statistical significance (for better or worse) is binary. Results approaching significance have to be reported as such. It would be useful for the authors to clarify this, since the melody/harmony results are quite clear.

Thank you for this key comment, which has led to the addition of two analyses to better introduce our results. First, we present a heat map of correlations between the behavioral measures (indices of synchronization accuracy and consistency) and the mean activity within the R55b, during each of the experimental conditions of each of the three musical paradigms (Figure 3). This map immediately exposes the behavior-brain association and its variation among paradigms and complexity levels. We refer to this variation in the Discussion (page 6, lines 258-263; page 7, lines 298-301). Second, we introduce a stepwise multiple linear regression analysis, where we included among the independent variables not only the R55b activity, but also mean activity within each of the remaining 35 parcels, which showed maximal activation during the Rhythm paradigm (Tables 2-3). This analysis now better clarifies how unique is the association between R55b activity and rhythmic entrainment metrics (Results, page 6, lines 239-247; Discussion, page 8, lines 322-340).

3) The discussion on the mirror neuron system is a bit weak. I would instead encourage the authors to not speculate here and instead stick with the well understood dorsal stream mechanisms.

We thank the reviewer for highlighting this point. In the revised manuscript, the reported associations: dorsal stream – sensorimotor integration and dorsal stream – rhythmic entrainment are explicitly mentioned in the Introduction (page 2, lines 47-48; page 3, lines 95-96), and the possible linkage of the R55b to the dorsal pathway is discussed (page 8, lines 342-347). Yet, we do not see a contradiction between the dorsal stream model and the mirror neuron system paradigm. The human mirror neuron network has been claimed to encompass not only ventral but also dorsal regions of the premotor cortex and the posterior parietal cortex [Cattaneo & Rizzolatti, 2009]. Mirror neurons have been implicated in music and language, prediction in complex hierarchical structures and sensorimotor associative learning [e.g., Molnar-Szakacs & Overy, 2006, Overy & Molnar-Szakacs, 2009, Kilner et al., 2007, Cook et al., 2014], therefore we believe that the mirror neuron system is highly relevant for the current discussion. The relevant paragraph has been updated accordingly (pages 8-9, lines 354-366).

Overall, this is a very interesting paper. With appropriate revisions, it could have a significant impact on the field of the biology of music perception in humans (and other animals).

Reviewer #2

Review for “The Rediscovered Motor-related Area 55b Emerges as a Core Hub of Music Processing”

General impression

This study provides useful insights on the specific role of R55b in music perception, and possibly generalizing to other functions, such as speech perception. The methods seem solid and rich, including both a meta-analysis and a targeted fMRI study. I particularly enjoyed the choice of naturalistic paradigm, in which participants listen to the stimuli without explicit task to focus on specific aspects of the stimuli. If the following comments are taken into account, mainly regarding further clarifications and connections to other functions, I would recommend acceptance for publication, as this study would contribute very valuable knowledge to the field.

We are grateful for the reviewer’s highly valuable and constructive remarks and for acknowledging the quality of our work.

Specific comments

Lines 74-75: “thus might represent the right homologue of the language hub advocated lately” this phrasing softly implies a left-right distinction between language and music. We indeed know that language is heavily left-lateralized, but especially speech comprehension is also distributed to both hemispheres and shares networks with other functions, such as auditory perception. So I would recommend rephrasing this sentence to not imply this binary hemispheric distinction, but to leave open the possibility that the right 55b might be also used in speech comprehension, as part of the auditory processing network (of course used for music as your study is focusing on).

We thank the reviewer for raising this key point. We have rephrased this sentence to avoid implication of simple binary hemispheric distinction, and cautiously mentioned a possible linkage with speech perception (page 7, lines 276-282).

Line 77: “probably”, is it not clear how many of the studies were testing the production, and how many studies tested the perception of music? Please provide the details, possibly in percentage of the total studies entering the meta-analysis.

In the new version of the manuscript we have added in the Results section the exact number of studies related to music perception, production, intervention/training, imagery and various combinations of the above (page 4, lines 139-143), as well as the percentage of studies focusing on music perception (page 4, line 154). Details of the 163 studies included in the meta-analysis are now provided in Supplementary Table 1.

Lines 87-88: “location 24, 32 and 29 in the rhythm, melody and harmony paradigms, respectively,” this is not clear as the format of the paper for this journal presents the results before the design. Please describe with words what were the designated locations, so that it is clear when one reads the paper serially.

The relevant paragraph of the Results has been expanded to better introduce the analysis presented in Figure 2 (page 4, lines 158-176).

Lines 91-111: This paragraph is not completely consistent, it starts with referring to motor regions, then touches upon the concepts of entrainment and synchronization, and quickly moves on to beat perception via prediction. I would recommend making it more coherent by omitting some of the details, and keeping only the necessary information for the point you need to make, as for example the phenomenon of negative asynchrony. Also please use a topic sentence (the first sentence of the paragraph) that states the argument you want to make in this paragraph and paves the way for the details provided next in the paragraph.

We completely agree with the reviewer. In the revised manuscript, this paragraph has been relocated to the Introduction and split into four paragraphs, which present more thoroughly the relevant current knowledge regarding the association between the auditory and motor systems (pages 2-3, lines 44-97).

Line 113: “rhythmic entrainment” please clarify whether you mean endogenous entrainment of the neural circuits or synchronization to beat as caused by external stimulus, or what exactly is the process you are assuming that your results track. (see for example: <https://journals.sagepub.com/doi/full/10.1177/0748730420972817> or a bit more relevant to the neural circuits: <https://www.frontiersin.org/articles/10.3389/fnhum.2018.00263/full> and [https://www.cell.com/trends/cognitive-sciences/fulltext/S1364-6613\(20\)30078-4](https://www.cell.com/trends/cognitive-sciences/fulltext/S1364-6613(20)30078-4)) There has been confusion in the field in the recent 2-3 years and new distinctions have emerged, also in the field of language processing, so it would be very useful to clarify with regard to music.

Thank you for this important comment. We have now defined the term “rhythmic entrainment” in the Introduction (page 2, lines 73-76), and provide evidence to support neural entrainment as the mechanism underlying motor rhythmic entrainment in the Discussion (pages 7-8, lines 283-294, 307-321).

Line 114: “59 subjects also performed a sensorimotor synchronization task outside the scanner”, did this happen before or after the fMRI measurements? If before, you need to acknowledge that there might be effects of this task on the passive task inside the scanner. I see now that this is in a separate session, maybe add a clarification in the sentence, stating that the fMRI measurements were done during the second session.

A clarification was added to the relevant paragraphs of the Results (page 5, lines 185-187) and Methods (page 20, lines 632-635) sections.

Lines 120-121: “the rhythm paradigm” this needs to be clarified, as the reader who reads linearly won’t know what the rhythm paradigm consists of. I would recommend to substitute this phrase with a short description of what the paradigm asked of the people, e.g. “passively following the rhythm of the music piece” or some similar phrase that describes the function of the paradigm, rather than simply naming it.

In the current version of the manuscript, the musical paradigms are described in more detail in the Results section (page 4, lines 158-166) and schematic illustrations are provided

(Supplementary Figures 1-3). Whenever possible, a short description of the task is presented instead of naming it.

Lines 132-133: “area 55b is proposed as an epicenter for perception-action coupling within a nexus of auditory-motor loops mediating sensorimotor integration and implicit prediction”, I like this definition, thank you. As you are being very specific (and well done) in this sentence, I would recommend adding one more sentence, possibly connecting to the more general functions of music perception and production, as well as speech perception and production. Would be nice to know what you propose with regard to these higher level functions, based on the specificity of this region. I see that you have a very interesting paragraph starting in line 144, but it would still be nice to add one sentence connecting to higher order function, else I’m left wondering about the very specific role without being able to generalize it to the functions I’m interested in.

The mentioned paragraph has been modified as part of the substantial revision of the Discussion. We hope the marked expansion of both the Introduction and Discussion, including the reference to music and language as higher order cognitive functions related to hierarchical prediction (page 2, lines 61-63) address this concern. We hypothesize that as a unimodal association region, area 55b, may take part in a medium complexity level processing loop along the mentioned hierarchical nexus of sensorimotor integration. However, this hypothesis is not yet well supported and extends beyond the scope of the current study.

Figure 1, panels A and B: It is clear from the coordinates at the figure caption that the yellow crosshair is in the right hemisphere, but I would also recommend adding the R and L letters on the corresponding sides of the template in panels A and B, adding visual clarity and making the figure able to almost stand alone.

R and L letters and MNI coordinates have been added in panels a and b of Figure 1.

Figure 2, lines 235-236 “to either rhythmic passages (N=67), instrumental melodies (N=66) or harmonic progressions (N=65)” it is not clear why there are different N in each contrast, didn’t all participants do all 3 paradigms? As you are referring to one study, I am assuming that all 71 participants were tested in all 3 paradigms in a block design? Please clarify if that is indeed the case and the length of each block in time. And then also please clarify the small deviations, why some participant’s data was discarded per paradigm, and was it the same participant for all three paradigms or not?

The reviewer’s understanding is correct. These important points regarding the sample sizes are now clarified in the Statistics and Reproducibility subsection of the Methods section (page 22, lines 737-747). Block and interval duration as well as the total time of each task are now also included in the Stimuli subsection of the Methods (pages 18-19, lines 572-575, 592-595, 606-609).

Line 297 “Participants”: You do not state the ethical approval for this study. This is very important information and needs to be added in the Participants section (unless I have missed something about Nature Communications regulations). If no ethical approval was obtained for

this study, I would not recommend publication.

Ethical approval information is now provided in more detail in the Participants subsection of the Methods (page 18, lines 545-548).

Lines 395-7, “Participants exhibiting head motion of >2 mm were excluded from analysis of the relevant task (four, five and six participants in the rhythmic, melodic and harmonic paradigms, respectively).” Please clarify why you excluded participants based on paradigm, instead of the whole participant, because this creates slightly different groups per condition and it is not clear why this is needed. I understand you do not compare stats between paradigms, which is good, given that you excluded datasets based on paradigm, but it would still be good to be transparent about how much overlap the different groups had, after exclusion of certain datasets per paradigm.

This issue is now covered in the first paragraph of the Statistics and Reproducibility subsection of the Methods, as mentioned above (page 22, lines 737-747). As for the overlap between groups following fMRI data exclusion, 58 of the 71 participants (i.e., 82%) had data for all three fMRI paradigms.

Lines 421-4 “For each paradigm we used the contrast between the condition of maximal enjoyment as documented by post-scan reports (rhythm: third level of complexity, melody: second level of complexity, harmony: regular cadence) and baseline, as maximal enjoyment is expected to characterize periods of maximal entrainment” This is not clear, what is the “condition of maximal entrainment”, how was this “documented”.

To better clarify this point in the current version of the manuscript we added a correlation matrix, which quantifies and visualizes the relationship between rhythmic entrainment indices and R55b activity in each of the experimental conditions (Figure 3). Maximal brain-behavior correlation was found for the third condition of the Rhythm paradigm, second condition of the Melody paradigm and the first condition of the Harmony paradigm. To avoid redundant analyses, multiple linear regressions were performed only for the conditions showing greater brain-behavior correlation. An equivalent pattern of variation in emotional ratings is also reported (Supplementary Table 2, Supplementary Figure 4), yet, the association between emotion and entrainment extends beyond the focus of the current work and therefore only shortly mentioned in the Discussion (page 7, lines 298-301).

Line 474 “accuracy (AA) ”: it is not clear to me why absolute asynchrony (AA) is a measurement of accuracy, please clarify this connection.

Here “accuracy” refers to how close in time the participant’s finger taps and the corresponding beat onsets are. AA is a measure of the mean linear time distance between the movements and the beat (page 5, lines 190-193) [See Dalla Bella et al., 2017]. AA served as a proxy of ‘negative asynchrony’ to allow simple statistical analyses. In our data, only 5 out of 59 participants (8%) showed mean positive asynchrony (median: 10.18 ms, range: 0.83 - 47.11 ms).

Lines 472-9 “Statistical analysis” Please provide a detailed description of the models you calculated and whether you compared different models or you just looked into the estimates of

the theoretically hypothesized model. It is not clear which functions of R you used, e.g. lmer(), anova(), summary() (?), and what the p values on the tables correspond to, effects inside each model, or differences between model fit?

The exact R functions used for each analysis are now detailed in the Statistics and Reproducibility subsection of the Methods section (pages 22-23, lines 753-770). The mentioned analysis did not include comparison between models, we were interested in the effect of R55b activity on each of the behavioral measures (AA, LRV-logit, ENT), while controlling for the covariates (age, gender and musical education). Accordingly, p values in Table 1 correspond to significance level of the F statistics of the hypothesized model. In addition, asterisks denote significance of the model's estimates.

Reviewer #3

Authors presented a manuscript on the pre-motor area 55b, suggesting it as a core hub of music processing. To do so, they computed a NeuroSynth meta-analysis, three original fMRI tasks and one behavioral study related to music perception.

Although the work may be of interest, the manuscript does not look adequate for the standard of Communications Biology. Indeed, introduction, results and discussion are mixed in a very short text describing previous literature and the current studies. Moreover, no statistical results are reported in the main text. They are shown only in Figures and Tables. I am not necessarily against this procedure, but it seems very far from the normal guidelines provided by Communications Biology. Thus, I would encourage authors to provide a more organized manuscript with a broader literature review and discussion of the results and a better presentation of their results. As follows, I report some guidelines offered by Communications Biology: <https://www.nature.com/documents/commsj-life-style-formatting-guide-accept.pdf>

We thank the reviewer for the constructive and helpful critique. This manuscript was originally submitted as a brief communication to Nature Neuroscience. It is now resubmitted following a thorough revision, in accordance with the guideline provided by Communications Biology. The Introduction, Results and Discussion have been expanded and now presented as independent chapters; statistical results are also reported in the main text.

If the editor is willing to allow authors to perform an in-depth revision of the manuscript based on the above points, here there are a few more technical comments that authors should address.

-Participants: it would be useful to have complete demographic information for the subsample of participants who took part in the behavioral study only. Moreover, the name of the "musical experience questionnaire" that authors used should be reported.

The complete demographic information for the subsample of participants who took part in the behavioral study is now provided in the Participants subsection of the Methods section (Page 18, lines 535-538). Our musical experience questionnaire was composed by Prof. Roni Granot to include a set of standard questions commonly used in formal questionnaires (e.g., years of formal musical education, listening hours per day, hours of instrument-playing per week, etc.) and a few questions referring to musical genres popular in Israel. However, for the current study we only used the information on years of formal musical education.

-fMRI tasks: I think authors did one fMRI study with three tasks and not three different fMRI tasks. Moreover, authors should better clarify what they mean by “music processing” since throughout the manuscript they talk about passive music listening, music perception, music entrainment, appreciation of music, etc. with regards to their analyses and choices. Along this line, authors used only the maximal enjoyed musical condition when computing the ROI analysis, claiming that this is connected to entrainment. Here, I would ask authors to show the results for all conditions (i.e. also for the ones where the music appreciation was low). Moreover, it seems that authors centered their manuscript on the concept of music perception/processing, while for the ROI analysis they are specifically talking about music entrainment or enjoyment of music, which is not the same thing.

We thank the reviewer for these helpful comments.

(1) The reviewer’s understanding is correct; we present results of one fMRI study with three tasks. The text was corrected accordingly (page 1, line 23; page 18, line 562).

(2) We have amended the text such that it now focuses on music perception. The term “music processing” is only used when referring to the NeuroSynth meta-analysis. Aspects of music processing dealt with in each of the meta-analysis studies are now detailed in Supplementary Table 1 and a summary is provided in the Results section (page 4, lines 139-142).

(3) As mentioned above, we now present the results of the correlation between measures of rhythmic entrainment and R55b activity for each of the experimental conditions of the three musical paradigms (Figure 3). The linkage between entrainment and perception is now better clarified in the Introduction (page 2, lines 79-81). In general, the renewed Introduction and Discussion expand on the association between music perception and the auditory-motor interaction, including the proposed mechanism of neural entrainment.

-NeuroSynth meta-analysis: authors must report more details on how they did the screening in the review phase. Apparently, they started from more than 14000 papers using only the word “music”. This does not sound like a good search, and it is not clear how they ended up having 163 studies. I would advise to do a systematic review following Prospero’s guidelines, and, even if they still want to do a non-systematic review, authors should at least provide more details on their procedure. Moreover, no details are present about the statistics used in the meta-analysis. This should be reported as well.

We thank the reviewer for this important comment. As mentioned earlier, in the current study, we used the NeuroSynth online database to derive meta-analysis of neuroimaging studies related to music. The purpose of this part of the study was to show that area 55b emerges as a hub of music processing, even when applying a coarse method of literature capturing. The meta-analytic map was downloaded from the NeuroSynth website and then overlaid on the HCP-MMP1 parcellation. We performed neither the screening and selection phases nor the statistical analysis of this meta-analytic study. However, in the current version of the manuscript the NeuroSynth term-based meta-analysis procedure is described in more detail (page 21, lines 685-701). Details of the 163 studies included in the meta-analysis are provided in Supplementary Table 1.

REVIEWERS' COMMENTS:

Reviewer #1 (Remarks to the Author):

The authors have addressed my comments.
I recommend acceptance.

-signed

Ramesh Balasubramaniam.

Reviewer #3 (Remarks to the Author):

Authors satisfactorily addressed the comments which I provided in my previous revision.
I am particularly glad to see more details on the procedures employed by the NeuroSynth meta-analysis which clarify my previous doubts.
I recommend the paper for publication.
However, I have the following minor suggestions that authors and editors may consider:
-the last paragraph of the Introduction should be removed since it describes results
-the Results section is really long and a few details may be removed and left only in the Methods.
However, for this last comment I would refer to the the recommendation made by the editors of Communication Biology